# Does Sex Matter in the Link Between Self-Efficacy and Mediterranean Diet Adherence in Adolescents? Insights from the EHDLA Study

**DOI:** 10.3390/nu17050880

**Published:** 2025-02-28

**Authors:** José Adrián Montenegro-Espinosa, José Francisco López-Gil

**Affiliations:** 1One Health Research Group, Universidad de las Américas, Quito 17024, Ecuador; josefranciscolopezgil@gmail.com; 2Department of Communication and Education, Universidad Loyola Andalucía, 41704 Seville, Spain

**Keywords:** dietary self-regulation, mediterranean dietary pattern, nutritional epidemiology, eating healthy, dietary habits, health behavior, psychological determinants, youths

## Abstract

**Background/Purpose:** To our knowledge, no previous study has analyzed the associations between self-efficacy and adherence to the Mediterranean diet (MedDiet) in adolescents, nor have sex-based differences in this relationship been examined. The aim of the current study was to examine the relationship between self-efficacy and MedDiet adherence in Spanish adolescents. **Methods:** This research was cross-sectional and involved 619 adolescents (56.5% girls) who were part of the Eating Habits and Activities of Daily Living (EHDLA) project. Self-efficacy was assessed using a 10-item scale (general self-efficacy scale), which yielded scores ranging from 20 to 100 points. The Mediterranean Diet Quality Index for Children and Adolescents (KIDMED), which consists of a 16-item questionnaire with scores ranging from −4 to 12, was used to evaluate adherence to the MedDiet. Unhealthy behaviors related to the MedDiet were assigned a score of −1, whereas healthy behaviors received a score of +1. **Results:** Overall, for each 20-point increase in self-efficacy, boys had a greater non-significant likelihood of having optimal MedDiet adherence (1.33%, 95% confidence interval [CI] −5.15 to 7.82, *p* = 0.687). Conversely, girls had a greater significant likelihood of having optimal MedDiet adherence per further point in self-efficacy (7.40%, 95% CI 2.28 to 12.53, *p* = 0.005). Individually, among boys, a 20-point increase in self-efficacy was associated with a 4.8% higher probability of having a dairy product for breakfast (95% CI 0.2 to 9.4, *p* = 0.042). Among girls, the same increase in self-efficacy was linked to a greater probability of consuming fruit or fruit juice daily (6.8%, 95% CI 2.2 to 11.4, *p* = 0.004), eating a second fruit every day (8.6%, 95% CI 3.4 to 13.8, *p* = 0.001), consuming fish regularly (6.2%, 95% CI 1.1 to 11.2, *p* = 0.017), enjoying pulses and eating them more than once a week (4.5%, 95% CI 0.3 to 8.8, *p* = 0.035), having cereals or grains for breakfast (5.4%, 95% CI 0.4 to 10.5, *p* = 0.035), and regularly consuming nuts (5.2%, 95% CI 0.2 to 10.2, *p* = 0.041). Additionally, increases in self-efficacy was associated with a 5.0% lower probability of skipping breakfast (95% CI −8.8 to −1.1, *p* = 0.012). **Conclusions:** Our results revealed a significant association between self-efficacy and MedDiet adherence among girls, whereas the relationship between self-efficacy and the MedDiet in boys was not significant. These results suggest that interventions aimed at improving adolescent dietary patterns should consider incorporating strategies to increase self-efficacy, potentially with sex-specific approaches.

## 1. Introduction

In recent years, the Mediterranean diet (MedDiet) has been recognized and recommended by various organizations worldwide, such as the European Nutrition for Health Alliance, the American Heart Association, and the World Health Organization, for its particular features and multiple health benefits [1,2,3]. Furthermore, scientific literature has shown an inverse relationship between adherence to the MedDiet and the incidence of noncommunicable diseases, such as malignant neoplasms, cerebrovascular accidents, cardiovascular pathologies, and diabetes mellitus [4]. The MedDiet is characterized by the abundant presence of whole grains, seasonal vegetable products (e.g., spinach, lettuce, kale, brussel sprouts), and fruit (e.g., grapes, figs, oranges, pomegranates, apples), with at least two to three portions daily, and the source of dietary fat is olive oil [5,6]. It also favors the consumption of fish at least twice a week over other animal protein or processed meats (e.g., pork meat, red meats, sausages, salami) and incorporates a measured intake of cheese, eggs, and milk [5,6]. In adolescents, adherence to the MedDiet increases high-density lipoprotein cholesterol levels, as well as reduces triglycerides, low-density lipoprotein cholesterol, and systolic blood pressure [7]. It also reduces body mass index and the percentage of obesity among children and adolescents [8].

A MedDiet during adolescence is associated with a lower risk of developing chronic diseases in adulthood, including metabolic syndrome, type 2 diabetes, and cardiovascular disease [9]. However, adherence to the MedDiet among adolescents is low, highlighting the need for educational initiatives to promote healthier eating habits from an early age [10]. Similarly, despite solid evidence supporting the benefits of the MedDiet, several systematic studies have reported a decrease in adherence in Mediterranean countries such as Spain, particularly among children and adolescents [10,11,12,13]. Research by Grosso & Galvano [10] evaluated several studies that were carried out in Spain between 2000 and 2015 to analyze adherence to the MedDiet in children and adolescents. These authors reported that, over time, adherence to the MedDiet decreased progressively, with a trend toward worse eating habits, resulting in high adherence in 2015, whereas 62% of Spanish children had low adherence [10]. A more recent study [13] similarly analyzed changes in adherence to the MedDiet among Spanish children and adolescents over two decades, using data from the enKid study (1998–2000) [14], and the Physical Activity, Sedentarism, Lifestyles and Obesity in Spanish Youth study (PASOS) (2019–2020) [15]. The results revealed a significant decline in MedDiet adherence over time. Specifically, the findings included lower consumption of fish (−20.3%), legumes (−19.4%), and fruits (−14.9%), along with higher consumption of pastries (+19.4) and fast food (+19.4%). Adolescents, especially those with secondary education, had the lowest adherence, at 10.9%. These results highlight the worsening dietary pattern among Spanish youth, emphasizing the need for specific interventions to promote healthier eating habits.

Self-efficacy refers to an individual’s belief in their ability to successfully carry out a behavior necessary to achieve a specific outcome [16]. A person’s expectations about his or her self-efficacy affect both the initiation and persistence of adaptive behavior in the face of challenging or novel situations [17]. Self-efficacy could be related to MedDiet adherence. First, self-efficacy, which is confidence in an individual’s ability to execute particular conduct, has been shown to directly influence healthy eating [18]. The greater the perceived ability to adhere to the MedDiet is, the better and healthier the individual’s eating behavior will be [19]. In addition, research suggests that self-efficacy may moderate the connection between food preferences and dietary intake [20]. This suggests that, even if a person prefers certain foods, his or her level of self-efficacy may determine his or her adherence to the MedDiet by influencing the decision to consume healthy foods or avoid those not recommended [21].

To our knowledge, no previous study has analyzed the associations between self-efficacy and adherence to the MedDiet in adolescents, nor have sex-based differences in this relationship been examined. However, existing research suggests that both self-efficacy and sex may play crucial roles in dietary behaviors. Fitzgerald et al. [22] analyzed the relationship between self-efficacy and food intake patterns in adolescents. Two distinct patterns were identified: “healthy food intake” (consumption of fruits and vegetables) and “unhealthy food intake” (consumption of snacks, cookies, and desserts). The results revealed that those participants with higher self-efficacy were directly related to healthy food intake and that this relationship was more prevalent in boys [22]. An intervention by Dzielka et al. [23], called Healthy Me, analyzed the importance of self-efficacy in eating behavior through an intervention with a sample of female adolescents divided into overweight and non-overweight, resulting in those with overweight and higher self-efficacy being more likely to maintain healthier eating behavior. Similarly, findings from the Family Life, Activity, Sun, Health, and Eating (FLASHE) study revealed that adolescent self-efficacy significantly mediates the relationship between perceived food-parenting practices and fruit and vegetable consumption, reinforcing its role in shaping adolescent’s eating behaviors [24]. With respect to the MedDiet, a study on adolescents with obesity revealed that girls had greater adherence to the MedDiet, reflected in a lower intake of saturated fats and greater consumption of fruits and vegetables [25]. Similarly, a systematic review [26] reported that during the Coronavirus Disease 2019 (COVID-19) pandemic, girls had greater adherence to the MedDiet than boys did, emphasizing the role of sex-specific social support.

Therefore, information on self-efficacy and adherence to healthier eating patterns or the MedDiet is not clearly defined by sex, and distinct elements need to be identified in this regard. Moreover, self-efficacy has been identified as a key determinant of dietary choices. This has led to the development of evaluation instruments, such as the Self-Efficacy Scale for Adherence to the Mediterranean Diet (SESAMeD), which was specifically developed and validated to assess how self-efficacy influences MedDiet adherence [18]. Given these findings, self-efficacy may influence MedDiet adherence differently in boys and girls. Thus, the present study aimed to examine the association between self-efficacy and adherence to the MedDiet in Spanish adolescents while also assessing the moderating role of sex in this relationship.

## 2. Methods

### 2.1. Study Design and Population

This secondary cross-sectional analysis utilized data from the Eating Habits and Daily Living Activities (EHDLA) study. The research framework employed in this study includes sample size determination as previously described, which has already been documented in the EHDLA study [27]. The sample comprised 619 adolescents (56.5% girls) aged 12–17 years who were recruited from three secondary schools in the *Valle de Ricote* (Region of Murcia, Spain) between 2021 and 2022. The inclusion criteria required participants to reside in or attend school in the specified region, with written consent from both guardians and adolescents. The exclusion criteria included the absence of physical education assessments, medical conditions contraindicating physical activity, active medical treatment, or lack of parental authorization.

Ethical approval was granted by the Ethics Committee of the Albacete University Hospital Complex and Albacete Integrated Care Management (ID 2021–85, approved on 23 November 2021) and the Bioethics Committee of the University of Murcia (ID 2218/2018, approved on 18 February 2019). The research adhered to the Declaration of Helsinki principles to preserve participant rights.

### 2.2. Measures

All assessments and questionnaires were conducted face-to-face by trained personnel from the research team during physical education classes throughout the academic period.

#### 2.2.1. Self-Efficacy (Independent Variable)

Self-efficacy was assessed via the validated Spanish version of the General Self-Efficacy Scale [28], which has high reliability (Cronbach’s α = 0.89) in prior adolescent populations [29]. The original 10-item instrument, which uses a 4-point Likert scale, was adapted to a 10-point Likert-type scale to ensure consistency with other instruments used in the research [30]. This instrument measures perceived competence in managing challenging situations (e.g., “I can find solutions to obstacles”) [25]. The total scores ranged from 10 to 100, with higher values indicating greater self-efficacy.

#### 2.2.2. Adherence to the Mediterranean Diet (Dependent Variable)

Adherence to the MedDiet was evaluated via the Mediterranean Diet Quality Index for Children and Teenagers (KIDMED), a 16-item questionnaire validated for Spanish youth [31]. The items were scored as −1 (unhealthy behaviors, e.g., fast food consumption) or +1 (healthy behaviors, e.g., fruit/vegetable intake), yielding total scores ranging from −4–12. Scores were categorized as ≤3 (very low adherence), 4–7 (requires improvement), or ≥8 (optimal adherence). For analytical purposes, categories were dichotomized into optimal (≥8) and nonoptimal (≤7).

#### 2.2.3. Covariates

Sex and age were self-reported by the adolescents. Socioeconomic status was collected via the Family Affluence Scale-III (FAS-III) [32]. The FAS-III composite score (0–13) quantified familial economic resources. Physical activity and sedentary behavior were assessed via the Youth Activity Profile Physical (YAP), a self-report questionnaire with 15 items [33]. The Spanish edition of the YAP was employed in this study (the YAP-S) [34] with a 5-point Likert scale, which assesses physical activity and sedentary behavior across three domains: school-based activity, out-of-school activity, and sedentary habits. Sleep duration was calculated as [(weekday sleep hours × 5) + (weekend sleep hours × 2)]/7, derived from self-reported bed and wake times by two questions, differentiating between weekdays and weekends: “At what time do you typically go to bed?” and “At what time do you usually wake up?”. A validated Spanish food frequency questionnaire (FFQ) [35] was used to estimate the daily energy intake (kcal) of each participant. Body weight was assessed via the Tanita BC-545 scale ± 0.1 kg, and height was measured with a Leicester Tanita HR 001 stadiometer ± 0.1 cm. Body mass index was computed as kg/m^2^. The participants were measured in lightweight clothing.

### 2.3. Statistical Analysis

Variable distributions were evaluated via Shapiro-Wilk test, quantile–quantile plots, and density plots. Continuous variables are presented as medians and interquartile ranges (IQRs), and categorical variables are presented as percentages. The primary analyses revealed a significant interaction effect between sex and self-efficacy scores concerning the KIDMED score (*p* = 0.019) and a trend towards significance for the probability of having an optimal MedDiet (*p* = 0.087). Consequently, both boys and girls were stratified and analyzed separately.

To examine the associations between self-efficacy scores and the KIDMED score or the likelihood of achieving an optimal MedDiet, generalized linear models (GLMs) were implemented. These analyses incorporated robust estimation techniques to address potential heteroscedasticity and outliers [36]. The estimated marginal means (M) and predictive probabilities (%), along with their corresponding 95% confidence intervals (CIs), were computed for the general self-efficacy scale on the basis of the KIDMED score and the likelihood of attaining an optimal MedDiet. To facilitate interpretation, these parameters were obtained for every 20-point increase in the self-efficacy scale rather than for each individual point. The results per single-point increase can be derived by dividing the parameters by 20, except for the *p* value, which remains constant.

The models were adjusted for several covariates, including age, sex, the FAS-III score, physical activity, sedentary behaviors, overall sleep duration, body mass index, and energy intake. Statistical analyses were performed via R statistical software version 4.3.2 (R Core Team, Vienna, Austria) in conjunction with RStudio version 2023.09.1+494 (Posit, Boston, MA, USA). A *p* value threshold of less than 0.05 was applied to determine statistical significance.

## 3. Results

Table 1 shows the descriptive data of the study participants. The median KIDMED score was 7.0 (IQR = 3.0) for boys and 6.0 (IQR = 3.0) for girls. The median general self-efficacy scale score was 74.5 (IQR = 24.0) for boys and 73.0 (IQR = 26.5) for girls.

Figure 1 displays the predictive margins of the KIDMED score for every 20-point increase in the self-efficacy-scale score. Among boys, the KIDMED score increased non-significantly with self-efficacy (unstandardized beta coefficient [*B*] = 0.01, 95% CI −0.01 to 0.02, *p* = 0.419). In contrast, among girls, the KIDMED score significantly increased with increasing self-efficacy (*B* = 0.03, 95% CI 0.02 to 0.04, *p* < 0.001). The full results of the GLM examining the relationship between self-efficacy-scale and KIDMED stratified by sex are found in the Appendix A (boys) and Appendix A (girls), respectively.

Figure 2 displays the predictive margins for self-efficacy on optimal adherence to the MedDiet. For each 20-point increase in self-efficacy, boys had a greater non-significant likelihood of having optimal MedDiet adherence (1.33%, 95% CI −5.15 to 7.82, *p* = 0.687). Conversely, girls had a greater significant likelihood of having optimal MedDiet adherence per further point in self-efficacy (7.40%, 95% CI 2.28 to 12.53, *p* = 0.005). The complete results of the GLM examining the relationship between self-efficacy-scale and adherence to the MedDiet stratified by sex are shown in the Appendix A (boys) and Appendix A (girls), respectively.

Table 2 presents the predictive probabilities of meeting each specific KIDMED item for every 20-point increase in the self-efficacy score. Among boys, a 20-point increase in self-efficacy was associated with a 4.8% greater probability of having a dairy product for breakfast (95% CI 0.2 to 9.4, *p* = 0.042). Among girls, the same increase in self-efficacy was linked to a greater probability of consuming fruit or fruit juice daily (6.8%, 95% CI 2.2 to 11.4, *p* = 0.004), eating a second fruit every day (8.6%, 95% CI 3.4 to 13.8, *p* = 0.001), consuming fish regularly (6.2%, 95% CI 1.1 to 11.2, *p* = 0.017), enjoying pulses and eating them more than once a week (4.5%, 95% CI 0.3 to 8.8, *p* = 0.035), having cereals or grains for breakfast (5.4%, 95% CI 0.4 to 10.5, *p* = 0.035), and regularly consuming nuts (5.2%, 95% CI 0.2 to 10.2, *p* = 0.041). Additionally, increases in self-efficacy was associated with a 5.0% lower probability of skipping breakfast (95% CI −8.8 to −1.1, *p* = 0.012).

## 4. Discussion

Our findings indicate that higher self-efficacy scores are associated with greater adherence to the MedDiet only among girls, whereas no significant association was observed in boys. Specifically, for each increase in self-efficacy, girls had a significantly higher likelihood of achieving optimal MedDiet adherence, while this association was not statistically significant in boys. These results suggest that self-efficacy may play a more relevant role in shaping dietary behaviors among girls.

One possible explanation is that boys in our sample exhibited higher baseline self-efficacy scores, which may have reduced the strength of the association with MedDiet adherence. Since their self-efficacy was already high, additional increases may not have led to substantial changes in dietary behaviors. In contrast, among girls, self-efficacy levels were slightly lower on average, and therefore, improvements in self-efficacy may have had a greater impact on their dietary choices. Fitzgerald et al. [22] found that boys with higher self-efficacy exhibited healthier eating habits, despite not specifically addressing MedDiet adherence. Moreover, the FLASHE study, which analyzed parent-adolescent dyads, found that self-efficacy mediated the association between food parenting practices and fruit and vegetable consumption, although adolescent characteristics, including sex, did not significantly moderate this mediation effect [24]. These results suggest that the influence of parenting practices on adolescent dietary behaviors may operate similarly across sexes rather than being stronger in girls as previously hypothesized. Nevertheless, our findings partially align with prior research indicating that girls tend to report higher self-efficacy regarding the selection and consumption of fruits and vegetables, which may contribute to their greater adherence to the MedDiet.

These findings provide further insight into how self-efficacy can promote healthier eating patterns in both boys and girls. However, the underlying mechanisms may differ between sexes, as different external factors appear to shape the influence of self-efficacy on dietary behaviors. Both Fitzgerald et al. [22] and Orlowski et al. [24] reported positive associations between higher self-efficacy levels and healthier eating habits. However, the factors reinforcing this association appear to differ between boys and girls. Fitzgerald et al. [22] found that peer support played a crucial role in promoting healthy eating among boys but not among girls. In contrast, Orlowski et al. [24] did not find significant sex-based differences in the mediation effect of self-efficacy on the relationship between food parenting practices and fruit and vegetable consumption. This suggests that while boys may require peer reinforcement to maintain healthy dietary habits, and girls may benefit more from parental support and structured food environments, the overall impact of parenting practices on dietary behaviors may not be inherently different between sexes.

Additionally, Dzielka et al. [23] demonstrated that interventions aimed at enhancing self-efficacy in girls led to positive changes in their eating behaviors over a three-month follow-up period. This finding supports the idea that self-efficacy may be a more critical determinant of dietary improvements among girls, particularly when reinforced by external factors such as parental influence. These differences highlight the importance of considering sex-specific approaches when designing interventions to improve dietary habits in adolescents.

Girls may be particularly receptive to interventions that reinforce their self-image and enhance their ability to manage changes in eating and physical activity habits, especially those with overweight. This aligns with the findings of Orlowski et al. [24], who reported that adolescent self-efficacy mediated the relationship between food parenting practices and fruit and vegetable consumption. However, their results did not indicate that this mediation effect varied by sex. Despite this, prior research suggests that girls tend to show greater interest and awareness in topics related to food and body image, which may strengthen their perceived control and ability to make healthy food choices independently. In contrast, boys may rely more on consistent reinforcement from peers to sustain healthy eating habits [22]. Although these studies do not specifically examine MedDiet adherence, they provide insight into the mechanisms through which self-efficacy may influence healthy eating behaviors and the potential sex differences in this relationship.

The predicted adherence scores for each specific KIDMED item per 20-point increase in self-efficacy may help explain the observed sex differences in our results. For instance, we found a significant association between higher self-efficacy and greater fruit consumption among girls. This aligns with findings from Helgadóttir et al. [37], who reported that girls tend to consume more fruit than boys. This difference may be influenced by specific upbringing patterns and the fact that girls generally mature earlier, shaping their dietary behaviors during adolescence.

Additionally, our results indicate that girls were less likely to skip breakfast, which further promotes daily consumption of fruits and cereals, whereas boys were more likely to skip this meal. This is consistent with previous research showing that girls score higher on the Breakfast Attitude Questionnaire (BAQ) and the Self-Efficacy for Healthy Eating Questionnaire (SEHE), particularly among those who do not skip breakfast [38]. Moreover, the same study found that boys scored higher on the ‘eating without awareness’ scale, suggesting that unconscious eating behaviors may negatively impact their healthy food choices.

The reasons behind these sex differences have been explored in previous studies analyzing biological, psychological, and sociocultural factors influencing nutritional behavior [39]. Research indicates that boys tend to consume higher-energy, high-fat diets due to their larger body structure, while girls generally prefer fiber-rich foods (e.g., fruits, vegetables, and cereals) and are more susceptible to social pressures related to weight and appearance [40]. As a result, girls may be more inclined—and perhaps more capable—of maintaining a healthier diet, whether by personal initiative or in response to social expectations.

These findings underscore the need for sex-specific strategies when designing nutritional interventions for adolescents. Future programs aimed at improving dietary habits should consider incorporating peer reinforcement strategies for boys, while recognizing the role of parental influence in fostering self-efficacy and healthy eating behaviors in both boys and girls.

This study’s limitations include its cross-sectional design, which precludes the establishment of causal relationships. Consequently, future research should employ longitudinal designs to assess whether higher self-efficacy genuinely enhances adherence to the MedDiet among adolescents. Additionally, the reliance on self-reported data introduces the potential for social desirability and recall biases.

Nevertheless, this study has notable strengths, as it represents, to our knowledge, the first investigation to verify the association between self-efficacy and MedDiet adherence in Spanish adolescents. It also provides cross-sectional evidence regarding the relationship of psychological with dietary factors within an understudied population (i.e., adolescents). Furthermore, the inclusion of multiple covariates, such as sociodemographic, anthropometric, and lifestyle variables, enhances the robustness of the findings. The study was conducted using a sample of adolescents from the *Valle de Ricote* (Region of Murcia, Spain), which contributes to achieving sufficient statistical power. Finally, this research is the first to explore the relationship between self-efficacy and MedDiet adherence specifically in Spanish adolescents, providing a foundation for future interventions tailored to this population.

## 5. Conclusions

Our results revealed a significant association between self-efficacy and MedDiet adherence among girls, whereas the relationship between self-efficacy and the MedDiet in boys was not significant. These results suggest that adolescents’ dietary patterns should consider incorporating strategies to increase self-efficacy, potentially with sex-specific approaches. Future research should explore the longitudinal relationships between self-efficacy and adherence to the MedDiet, as well as the effectiveness of self-efficacy-based interventions in promoting healthy eating patterns among adolescents.

## Figures and Tables

**Figure 1 nutrients-17-00880-f001:**
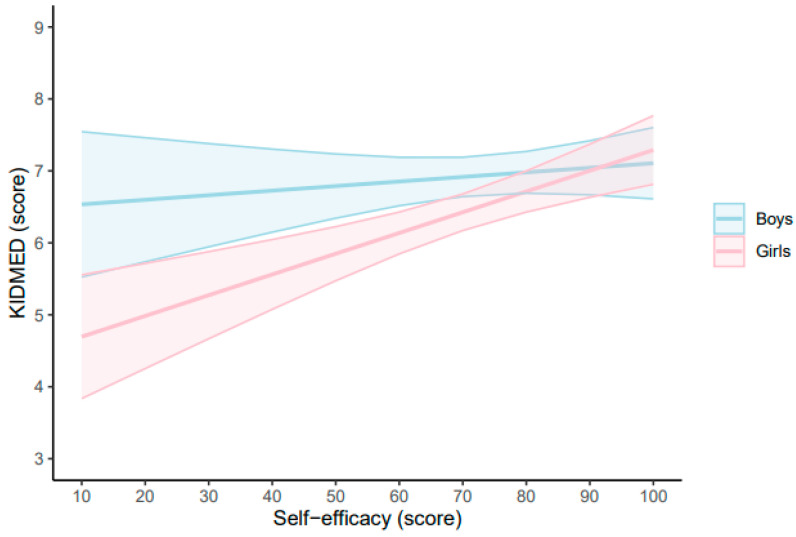
Estimated marginal means of the general self-efficacy scale based on the Mediterranean Diet Quality Index in children and adolescents scores in boys and girls. General self-efficacy scale adjusted for age, sex, socioeconomic status, physical activity, sedentary behavior, sleep duration, body mass index, and energy intake. KIDMED, Mediterranean Diet Quality Index in children and adolescents.

**Figure 2 nutrients-17-00880-f002:**
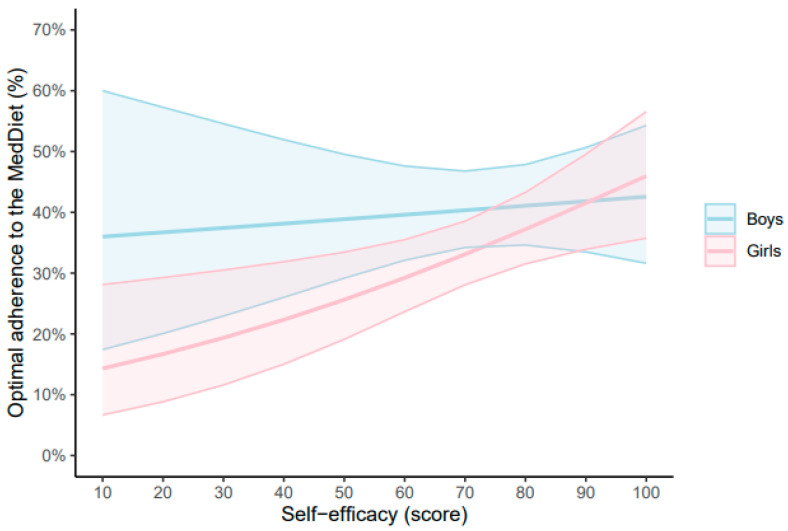
Estimated marginal means of the general self-efficacy scale based on optimal adherence to the MedDiet in boys and girls. General self-efficacy scale adjusted for age, sex, socioeconomic status, physical activity, sedentary behavior, sleep duration, body mass index, and energy intake. MedDiet, Mediterranean diet.

**Table 1 nutrients-17-00880-t001:** Descriptive data of the study participants.

Variables	Boys (n = 269; 43.5%)	Girls (n = 350; 56.5%)	Total (n = 619; 100%)
Age (years)	14.0 (2.0)	14.0 (2.0)	14.0 (2.0)
FAS-III (score)	8.0 (2.0)	8.0 (2.0)	8.0 (2.0)
YAP-S physical activity (score)	2.7 (0.8)	2.6 (0.9)	2.6 (0.9)
YAP-S sedentary behaviors (score)	2.6 (1.0)	2.4 (0.8)	2.6 (0.8)
Overall sleep duration (minutes)	505.7 (68.6)	492.9 (72.9)	501.4 (72.9)
Energy intake (kcal)	2576.6 (1446.2)	2637.6 (1508.9)	2631.9 (1494.5)
BMI (kg/m^2^)	22.1 (6.5)	21.5 (5.5)	21.7 (5.7)
KIDMED (score)	7.0 (3.0)	6.0 (3.0)	7.0 (3.0)
Adherence to the MedDiet			
Nonoptimal	160 (58.4)	236 (65.0)	236 (38.1)
Optimal	114 (41.6)	127 (35.0)	383 (61.9)
Self-efficacy (score)	74.5 (24.0)	73.0 (26.5)	74.0 (26.0)

Data expressed as median (interquartile range) or numbers (percentage). BMI, body mass index; FAS-III; Family Affluence Scale-III; KIDMED, Mediterranean Diet Quality Index for children and adolescents; MedDiet, Mediterranean diet; YAP-S, Spanish Youth Active Profile.

**Table 2 nutrients-17-00880-t002:** Predictive probabilities of meeting each specific item of the Mediterranean Diet Quality Index in children and adolescents per additional point in the self-efficacy score.

Self-Efficacy (per Each 20 Points) (Predictor)				
	Boys		Girls	
KIDMED Items (Outcomes)	% (95% CI)	*p* Value	% (95% CI)	*p* Value
Item 1: Takes a fruit or fruit juice every day	−0.2 (−5.9 to 5.4)	0.933	6.8 (2.2 to 11.4)	**0.004**
Item 2: Has a second fruit every day	0.3 (−6.3 to 6.9)	0.934	8.6 (3.4 to 13.8)	**0.001**
Item 3: Has fresh or cooked vegetables regularly once a day	1.4 (−4.5 to 7.4)	0.638	−3.3 (−8.2 to 1.5)	0.180
Item 4: Has fresh or cooked vegetables more than once a day	−0.9 (−7.0 to 5.3)	0.784	−0.8 (−6.0 to 4.4)	0.754
Item 5: Consumes fish regularly (at least 2–3 times per week)	−0.2 (6.7, 6.3)	0.949	6.2 (1.1 to 11.2)	**0.017**
Item 6: Goes more than once a week to a fast-food (hamburger) restaurant	1.7 (−4.2 to 7.6)	0.567	2.1 (−2.9 to 7.2)	0.414
Item 7: Likes pulses and eats them more than once a week	3.8 (−1.9 to 9.5)	0.195	4.5 (0.3 to 8.8)	**0.035**
Item 8: Consumes pasta or rice almost every day (5 or more times per week)	−2.7 (−9.2 to 3.9)	0.425	3.5 (−1.4 to 8.5)	0.162
Item 9: Has cereals or grains (bread, etc.) for breakfast	2.3 (−4.2 to 8.8)	0.488	5.4 (0.4 to 10.5)	**0.035**
Item 10: Consumes nuts regularly (at least 2–3 times per week)	4.9 (−1.6 to 11.4)	0.142	5.2 (0.2 to 10.2)	**0.041**
Item 11: Uses olive oil at home	2.0 (−0.3 to 4.3)	0.092	0.5 (−0.9 to 1.8)	0.507
Item 12: Skips breakfast	−2.4 (−6.0 to 1.2)	0.192	−5.0 (−8.8 to −1.1)	**0.012**
Item 13: Has a dairy product for breakfast (yogurt, milk, etc.)	4.8 (0.2 to 9.4)	**0.042**	4.4 (−0.3 to 9.1)	0.066
Item 14: Has commercially baked goods or pastries for breakfast	−1.2 (−6.0 to −3.7)	0.635	0.7 (−3.1 to 4.5)	0.707
Item 15: Takes two yogurts and/or some cheese (40 g) daily	−2.8 (−9.2 to 3.6)	0.390	4.2 (−1.1 to 9.4)	0.123
Item 16: Takes sweets and candy several times every day	−1.4 (−6.3 to 3.5)	0.577	−2.9 (−7.4 to −1.6)	0.207

Adjusted for age, sex, socioeconomic status, physical activity, sedentary behavior, sleep duration, energy intake, and body mass index. CI, confidence interval. KIDMED, Mediterranean Diet Quality Index in children and adolescents. Bold indicates a *p* value below 0.05.

## Data Availability

The data used in this study are available upon request from the corresponding authors. However, given that the participants are minors, privacy and confidentiality must be respected.

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
