# Peer review of "Does Sex Matter in the Link Between Self-Efficacy and Mediterranean Diet Adherence in Adolescents? Insights from the EHDLA Study"

_nutrients, 2025, doi:10.3390/nu17050880_

Round 1
Reviewer 1 Report
Comments and Suggestions for Authors
Dear Authors,
I thank the Editor for entrusting me to review this manuscript. Proper eating habits are extremely important in everyone's diet. This issue becomes even more important when it comes to young people (teenagers). Improper nutrition in childhood or adolescence can cause various health consequences in adulthood. One of the recommended diets is the Mediterranean Diet (MedDiet), which is considered beneficial to human health. Therefore, I congratulate the authors of this study for undertaking a study on Spanish adolescents to evaluate the relationship between self-efficacy and MedDiet adherence.
Below are my suggestions / comments:
Introduction section developed rather vaguely.
According to the information provided in the discussion of the results, the survey was conducted with a representative sample. By what characteristics was the sample representative? No such information is available in the methods section.
Why was a 10-point Likert scale used in the General Self-Efficacy Scale, while the original scale is a 4-point scale? Is it a Likert Scale at all?
Figure 1 and 2 Predictive margins for self-efficacy on the KIDMED scale are shown. However, for boys the unstandardized beta coefficient [B] is not statistically significant.
Standardize the notation of the literature item in the text (sometimes it is written after the space, sometimes together with the text).
Author Response
Please, see the attachment.

Reviewer 2 Report
Comments and Suggestions for Authors
In the abstract, before presenting the study’s objective you have to include a background statement. What is mentioned in Methods should be in Results (Line 11). Indications regarding future studies to be conducted should be pointed out.
“eating health” is not an appropriate keyword. The English language needs professional revision in the whole manuscript.
References must be formatted according to the journal’s instructions.
The Introduction needs to be expanded and improved. Especially, more data on self-efficacy and Mediterranean Diet should be provided.
The sample characteristics shouldn’t be in Methods, please move them to the Results section.
The ethical approval date should be given.
All the procedures have to be better explained in detail.
In the Discussion, more studies should be cited and their results be analyzed in comparison to yours.
The Conclusions are adequate.
Comments on the Quality of English Language“eating health” is not an appropriate keyword. The English language needs professional revision in the whole manuscript.
Author Response
Please, see the attachment.

Reviewer 3 Report
Comments and Suggestions for Authors
“Does Sex Matter in the Link between Self-Efficacy and Mediterranean Diet Adherence in Adolescents? Insights from the EHDLA Study”( nutrients-3483240)
This manuscript aimed to explore the potential sex differences in the link between self-efficacy and Mediterranean Diet Adherence in Spanish adolescents from the re-analysis the data from the EHDLA Study. The results revealed a significant association between self-efficacy and MedDiet adherence among girls, whereas the relationship between self-efficacy and the MedDiet in boys was not significant. Overall, this topic is interesting and important. However, some concerns appeared after reading the whole manuscript.
- The literature review part is far from satisfactory.
Some important papers need to be reviewed and discussed, such as,
Gomez Sanchez, M., Gomez Sanchez, L., Patino-Alonso, M. C., Alonso-Domínguez, R., Sánchez-Aguadero, N., Lugones-Sánchez, C., ... & Gomez-Marcos, M. A. (2020). Adherence to the Mediterranean diet in Spanish population and its relationship with early vascular aging according to sex and age: EVA study. Nutrients, 12(4), 1025.
Di Renzo, L., Gualtieri, P., Frank, G., Cianci, R., Raffaelli, G., Peluso, D., ... & De Lorenzo, A. (2024). Sex-Specific Adherence to the Mediterranean Diet in Obese Individuals. Nutrients, 16(18), 3076.
Brandt, G., Pahlenkemper, M., Ballero Reque, C., Sabel, L., Zaiser, C., Laskowski, N. M., & Paslakis, G. (2025). Gender and sex differences in adherence to a Mediterranean diet and associated factors during the COVID-19 pandemic: a systematic review. Frontiers in Nutrition, 11, 1501646.
- Many directly-related references mentioned in the discussion part should be reviewed in the introduction part first to help the readers get better understanding of state of art of the current topic.
- The sex differences on adherence to a Mediterranean diet has been validated in previous studies already, so is the relationship between self-efficacy and Mediterranean Diet Adherence, then why the current investigation is still needed? Why did you expect there might be difference or not in Spain?
- “2.2. Procedures” should be “2.2. Measures”. For procedure, you should mention how did you collect the data, by online questionnaires or face-to-face interview.
- I strongly recommend that the paper be thoroughly proofread and edited for languages and grammars, to enhance readership.
Comments on the Quality of English Language
The English could be improved to more clearly express the research.
Author Response
Please, see the attachment.

Reviewer 4 Report
Comments and Suggestions for Authors
Dear authors, I read your article, Please give us more informations about the food:
-fruits - what kind of fruits, what quantity was eaten daily
- meat, fast-food, etc
- they drink water, limonade, coke, etc? What quantity, how often?
- how many minutes make sport on a day?
Author Response
Please, see the attachment.

Round 2
Reviewer 3 Report
Comments and Suggestions for Authors
Thanks for the revisions and no further concerns.